# A Novel Framework for Generating Personalized Network Datasets for NIDS Based on Traffic Aggregation

**DOI:** 10.3390/s22051847

**Published:** 2022-02-26

**Authors:** Pablo Velarde-Alvarado, Hugo Gonzalez, Rafael Martínez-Peláez, Luis J. Mena, Alberto Ochoa-Brust, Efraín Moreno-García, Vanessa G. Félix, Rodolfo Ostos

**Affiliations:** 1Unidad Académica de Ciencias Básicas e Ingenierías, Universidad Autónoma de Nayarit, Tepic 63000, Mexico; pvelarde@uan.edu.mx; 2Academia de Tecnologías de la Información y Telemática, Universidad Politécnica de San Luis Potosí, San Luis Potosí 78363, Mexico; hugo.gonzalez@upslp.edu.mx; 3Facultad de Ingenierías y Tecnologías, Universidad De La Salle Bajío, Av. Universidad 602, León 37150, Mexico; rmartinezp@delasalle.edu.mx; 4Unidad Académica de Computación, Universidad Politécnica de Sinaloa, Ctra. Libre Mazatlán Higueras Km 3, Mazatlán 82199, Mexico; vfelix@upsin.edu.mx (V.G.F.); rostos@upsin.edu.mx (R.O.); 5Facultad de Ingeniería Mecánica y Eléctrica, Universidad de Colima, Av. Universidad 333, Colima 28040, Mexico; aochoa@ucol.mx; 6Dirección de Posgrado e investigación, Instituto Tecnológico de Tepic, Tepic 63175, Mexico; emoreno@ittepic.edu.mx

**Keywords:** intrusion detection, network security, traffic generation, machine learning, unbalanced dataset, botnet detection

## Abstract

In this paper, we addressed the problem of dataset scarcity for the task of network intrusion detection. Our main contribution was to develop a framework that provides a complete process for generating network traffic datasets based on the aggregation of real network traces. In addition, we proposed a set of tools for attribute extraction and labeling of traffic sessions. A new dataset with botnet network traffic was generated by the framework to assess our proposed method with machine learning algorithms suitable for unbalanced data. The performance of the classifiers was evaluated in terms of macro-averages of *F*1-score (0.97) and the Matthews Correlation Coefficient (0.94), showing a good overall performance average.

## 1. Introduction and Motivation

Nowadays, cybersecurity plays a fundamental role to ensure the usability and integrity of the information technology and telecommunication infrastructure. These technologies are fundamental for the common activities of organizations, enterprises, and individuals. As an example, using perimeter security it is possible to implement a layered approach protection, with the objective to identify and stop cyber attacks or network anomalies in the inbound and outbound network traffic using network monitor techniques over a network segment. In recent years, Network Intrusion Detection Systems (NIDS), based on anomalies (Anomay-based NIDS), had been incorporated machine learning (ML) and deep learning models to detect malicious network traffic patterns with excellent results [1,2].

NIDS and Intrusion Prevention Systems (IPS) are part of the defense strategies from cybercriminal tactics and attacks. However, this task to be efficient requires some desirable features in the NIDS, as such as: (1) fault tolerant; (2) minimum of human intervention on the administration of the devices; (3) avoid excessive work over system resources; (4) detect significant deviations from acceptable behavior; (5) high precision to minimize false positives and false negatives; (6) detect all kinds of patterns and sophisticated attacks; and (7) quick to detect intrusions and response effective to reduce possible damages [3,4].

To accomplish these desirable features on a NIDS, its necessary a large and comprehensive experimentation and modeling work, where most of the available conditions of work for a NIDS be considered. Usually, those conditions are represented through datasets. Furthermore, the features of these datasets determined the final characteristics of the NIDS.

Several works on NIDS study field have been presented through the years, all of them with the common element of requiring at least one dataset. Over time, several public network datasets had emerged, these often have their heyday and are subsequently being displaced by updated versions or new datasets with better features. As an example, the first synthetic labeled network datasets were produced from simulations of network scenarios that captured normal activity and a limited number of malicious actions. The synthetic nature of the datasets imposes some limitations by not capturing conditions that occur in a real environment, such as temporal patterns, frequencies, and work cycles. On the other hand, datasets ageing also affects them. These become obsolete to the lack of the latest attacks trend. Recently, new dynamic and synthetic datasets based on specific profiles have been developed to allow the creation of network scenarios with different categories of attacks, and that mimic the behavior of users to represent legitimate activity [5,6]. However, despite significant efforts to create better datasets, we consider that there are still open problems, and consequently represent areas of opportunity to improve:1.The time between when a dataset is created and updated is quite long, which is a critical situation given the speed with which evolves the complexity and the number of attacks;2.The produced datasets are not accurate in terms of real environments when these are created using simulations, since these present the behavior models related to users and attacks that were defined by the dataset designer;3.The datasets generated from a production network usually does not include full network dumps due to privacy issues and sensible content. In practices, from these datasets only files with features extracted by diverse tools, such as Argus [7], are shared in the form of CSV files. Without the full network dump is not possible to define other features at packet or network flow level.4.There is a lack of tools on the research community for off-line labeling of the network dumps. The available tools only extract network traffic features without offering a classification label. Thus, there is still a strong dependency on the current public datasets.5.The available tools for feature extraction on network traffic have some efficiency problems on files over 100 MB of size [8]. The feature extraction occurs at session level, thus, if one network session is not complete with all the packets that are part of it, the statistics are imprecise. We could verify that some tools do not check for the full network session.

The main motivation of this work is to solve the actual limitations of public datasets, which started to be used by the research community more than twenty years ago. Those datasets are considered as a static alternative, because they become obsolete as they do not reflect current threats. Having only this alternative limit the possibility to study other specific attack scenarios on production networks. On the other hand, personalized and dynamic datasets generated from real world network traffic and actual malicious traffic help to improve the state of the art and allow to use ML techniques for developing better NIDSs that fight the increasing cybercrime threats.

Therefore, in this paper, we propose a novel framework to generate personalized network datasets to help address the limitations presented before. In this sense, the proposed framework can be used as a suitable option to generate datasets that include specific attack scenarios over a production network, without the attack really happen. Thus, our proposal is three-fold:For the obsolescence and updates on network traffic datasets we propose the use of public repositories with real network malicious traffic, such as malware-traffic-analysis.net, which is updated daily with network dumps and malware samples;For the lack of real environment scenarios, we propose a dynamic and extensible framework to generate network traffic datasets based in the off-line aggregation of malicious network traffic to the traffic for a production network;To process the network traffic dumps generated by our framework, we propose novel and efficient tools for the feature extraction and label assignment for the network traffic sessions.

In summary our contributions are aligned with the main proposal.

1.We developed a tool to generate network traffic from real traces, sanitized and mixed with malware traffic. That can produce different realistic scenarios;2.Another tool will take the generated PCAP file and produce a list of features and labels that can be used by ML techniques;3.We generate a network dataset with botnet action on it and perform a deep analysis using ML techniques, discovering a relation between macro average of F1 and MCC.

The rest of the paper is organized as follows: Section 2 offers a revision of the related work. Section 3 presents the framework architecture and describe it in detail. Dataset, evaluation metrics, experiments, and results from this work are presented in Section 4. Finally, conclusions for this paper are in Section 5.

## 2. Related Work

The research field of network intrusion detection systems had been invigorated with the application of artificial intelligence techniques for intrusion detection. However, the main difficulty still remains: the scarcity of public network traffic datasets with the right features for the task. Several efforts had addressed to provide the research community with network datasets. There are some datasets that are considered emblematic in the evaluation of NIDS.

The oldest and most popular dataset is DARPA 1998. This dataset was generated by MIT Lincoln Laboratory with DARPA funding. From 1998 to 2001, this laboratory developed a series of simulations at big scale to create these synthetic data. MIT make them available to the public and can be downloaded from their website [9]. This dataset is composed by a training and testing parts, which include, respectively, seven and two weeks of network traffic. This dataset includes five class patterns: Normal, Probing Attack (Probe), Remote to Local (R2L), Denial of Service (DoS), and User to Root (U2R). However, there are some negative aspects [10,11], such as: the models used for network traffic generator are very simple and obsolete and do not represent the distribution of the network attacks in real scenarios (as an example, 79.2% of DoS versus 19.7% of normal traffic and only 1% for the rest of attacks). In 1999, a new improved version of the dataset was released as DARPA 1999. This dataset contains five weeks of network traffic, only the second includes a selected amount of attacks from its predecessor, it includes 200 instances of 56 different network attacks [12]. Because of their limitations, these datasets are no longer used in actual research works [13].

On the other hand, the KDD dataset is a family of synthetic datasets that includes KDD CUP 99 and NSL-KDD. The first [14] is based on DARPA dataset and contains around five million instances, each one representing a TCP/IP session made up of 42 features. It contains 22 attacks grouped into four categories (DoS, Probe, U2R, and R2L) and presents a class imbalance problem, since approximately 20% of them were classified as normal traffic patterns. Similarly, the problem is presented also in the attacks part; for example, U2R represents 0.15% in the training set and 0.7% in the test set. At the opposite extreme, DoS has a distribution of 79.2% in the training set and 73.9% in the testing set. Another problem is the huge number of redundant records in the training set and duplicated records in the test set, which negatively impacts ML algorithms [15]. The second dataset [6] is an improved and corrected version of KDD CUP 99 produced by the University of New Brunswick. It is made up of two training subsets and one test subset. The distribution of normal and attack traffic is nearly balanced in the three subsets. However, the distribution of the attacks is strongly skewed, more than 30% of the attacks are DoS and U2R does not reach even 1%. One of the improvements was the elimination of duplicate instances about 78% and 75% in the training and test sets, respectively.

The Kyoto 2006+ dataset was built with three years of real network traffic (from November 2006 to August 2009) using honeypots, darknet sensors, an email server, and web crawlers from the University of Kyoto [16]. Kyoto 2006+ dataset contains 14 statistical features derived from the KDD Cup 99 dataset and 10 additional features. This dataset was prepared to eliminate redundant instances and irrelevant features. One of its disadvantages is that there is no specific information about the attacks it contains.

Another recognized dataset is ISCX 2012, which was created by the University of New Brunswick in 2012 [5]. It is considered a dynamic one, because synthetic traffic is organized into a series of profiles that allow its selective use for different detection tests. The alpha profile describes the attack scenarios, these are used to generate malicious traffic for the evaluation of the systems. The dataset presents four scenarios: network infiltration from the inside, denial of services via HTTP, distribute denial of service via an IRC botnet, and brute force SSH attack.

The CTU-13 dataset [17] was captured from a production network at the Czech Technical University (CTU) in 2011 and is part of the Stratosphere IPS project [18]. The CTU-13 dataset consists of a group of 13 different malware captures made in a real network environment. These network captures include malware, normal and background traffic samples. The normal part of the dataset was captured using a Windows 7 running in a virtual machine. Normal traffic uses known web browsers, and the malware probably uses its own libraries to communicate with the Internet. Because it contains real network traffic, the full network traces are not publicly available.

All previous described datasets are now considered as static alternatives because of their features, since they do not change over time during the lifespan. Recently, new solutions to public datasets limitations had been proposed. As an example, new software tools had emerged, such as Silk [19]. The authors proposed a method to produce network traffic datasets for NIDS research by extracting meta-information from the network packets, combined with the logs from a IDS to label the traffic. The result generates labeled network flows compatibles with Netflow. That work allows to obtain access to a recent network flow datasets with associated labels. However, this proposal does not generate PCAP files, it only generates flow data, i.e., text files with a summary of specific features. Because it does not offer PCAP files, payload is missing and there are no possibilities to generate new and improved features. Additionally, the authors do not mention the IDS software that was used.

Other work proposed the software tool ID2T [20] to generate datasets for NIDS by injecting synthetic attacks that are mixed with background traffic. This approach produces files in both PCAP and XML formats, the latter used for labeling. The architecture has the ability to inject new and advanced attacks. However, it does not have the ability to guarantee that background traffic is attack-free, that is, it lacks of a traffic sanitization stage. With this, there is a risk of generating performance traces with unknown malicious traffic that will affect the training of a ML model. Another limitation of the work is that it does not support IPv6.

Wilaux and Ngamsuriyaroj [21] proposed a framework to generate bidirectional flow data with 20 features based on the combination of background and malicious traffic. For background traffic, they use 15-min captures of real traffic from the MawiLab project [22]. Then, they model the statistical properties of the traffic and, based on the adjusted model, they generate synthetic traffic using D-ITG (Distributed Internet Traffic Generator) [23], to use it as background traffic. On the other hand, malicious traffic is generated by tools that are part of the Kali Linux distribution [24]. Some issues with this proposal are the way they generate the background traffic, since the models used are memory-less, they cannot represent the self-similarity and long-term dependence properties of the real traffic [25]. MawiLab traffic is sanitized in a simplistic way, they only eliminate sessions of a packet and 0 milliseconds in duration, because they related it to scanning activity.

We identified some gaps in previous works related to the tools and data used to generate the synthetic traffic. For example, when using statistical models, the generated traffic does not adjust to a real network traffic. Additionally, the sanitization on MawiLab traffic is very simplistic and in some proposals the PCAP files are not available, while malicious traffic generated with kali Linux tools are synthetic. Another weakness is the output format for one tool, that only generates traffic in a NetFlow compatible format, discarding any payload and limiting the features that can be used to analyze that traffic. Our proposal framework includes the capability of mixing real and sanitized network traffic and real malicious traffic to generate PCAP files as datasets to investigate features and patterns. Our tool that extracts features supports IPv6. Additionally, traffic labeling can not only be performed to identify malicious and benign sessions, but each malicious session can also be labeled with one of the 13 priority 1 attack classifications defined by Snort in /etc/snort/classification.config.

## 3. Framework Architecture

In this section, we present the general architecture of the proposed framework. First, we offer an overview of architecture, which will be followed by a description of the phases that create it. Likewise, the production network used in this work is described.

The architecture proposed in this article is shown in Figure 1. The workflow consists of four phases. The first phase is data collection and cleansing, in which raw traffic from a production network is captured. The cleansing process takes a raw network trace in pcap format and removes the session packets that correspond to an alert generated by the Snort software [26]. Once all the alerts have been processed, a new attack-free network traffic trace according to Snort is obtained. This trace is also referred as sanitized trace. In the second phase, the aggregation of malicious traffic is carried out by replaying the sanitized trace and a trace of malicious traffic. In the third phase, the dataset is created in CSV format, the aggregated trace is processed to extract the features of the traffic sessions. Additionally, Snort software is used as an expert to help tag sessions as either attack or normal traffic. Optionally, with malicious traffic, those sessions could be sub-classified as one of the 38 attack types defined by Snort in the file classification.config. With this information, one more variable can be included with the type of attack provided by Snort. Finally, in phase four, the dataset is used by ML algorithms to select a predictive model that detects malicious traffic sessions. The details of each phase are explained in the following subsections.

### 3.1. Recollection and Cleansing Phase

As can be seen in Figure 2, this phase is made up of two parts, collection and cleansing. In the first part of this phase, network traffic is captured with TCPDUMP software, scheduling the capture task with cron at a set time. Real traces used in this research work were captured in the network of the campus of the Autonomous University of Nayarit (UAN). The following is a description of the network.

#### 3.1.1. Production Network Description

Figure 2 shows the architecture of the UAN network. It is a network with wired and wireless connections with approximately 7000 internal nodes organized into segments generated through the use of VLANs. These VLANs efficiently separate traffic and allow better use of resources through the logical segmentation of the infrastructure in different subnets. Hence, the packets switch only between ports within the same virtual network. The ER16 multilayer switch is responsible for directing traffic to the FORTIGATE 1000 appliance, and to the Demilitarized Zone (DMZ). The DMZ contains the WEB, DNS, EMAIL, and SQL servers. When a LAN node of the UAN requests access to an external host, the ER16 directs this request to the FORTIGATE 1000, which is in charge of managing the privileges, restrictions, and the priority of the traffic outside the UAN network. The capture sensor has an IP address of 192.100.162.12, and it is inside the DMZ on one of the email servers.

#### 3.1.2. Network Traffic Cleansing

Figure 3 shows the diagram where the collection of traffic and filtering of malicious sessions is carried out. A traffic session is defined as a set of packets that share a five-tuple: source IP address, destination IP address, source port, destination port, and protocol. The cleansing operation consists of eliminating all sessions that are involved with malicious activity. This process is automated and must be executed by an expert, for this reason was proposed the software Snort version 2.9.17 GRE (Build 199) was proposed. Snort is an open source signature-based IDS/IPS. It uses rules to define malicious activity in network traffic and based on them, it searches for matches in the monitored traffic, if that happens an alert is triggered. Alerts are stored in /var/log/snort/ and can be written in two types of binary format: PCAP dump or unified2. The latter is an extensible format formed by a header that defines the type of record and its length, and stores 21 types of information about the event, such as source and destination IP address, source and destination port, alert priority, classification of the attack, among others [27]. The alert file is used by our tool splitraffic, written in C++ to perform the cleansing operation on the raw trace, that is, to eliminate the traffic sessions that correspond to the alert. In the case of a PCAP alert file, the mapping is performed only on the five-tuple basis. Meanwhile, with unified2 alerts, in addition to considering the five-tuple information, we can have greater control over the sessions has to be eliminated. Specifically, network sessions can be removed based on the three priority levels defined in the file /etc/snort/classification.config by Snort. A priority alert 1 is assigned to the highest severity, 2 to the intermediate, and 3 to the lowest severity. The control of filtering using priorities is completed with the max_priority option, whose entered value represents the maximum number of alert priority considered to filter, i.e., max_priority = 1, it only takes into account priority 1 alert, max_priority = 2, it will use priority 1 and 2 alerts and similarly for level 3. With level 3, more alerts will be generated for priority 1, 2, and 3, which could lead to some false positives. For this work, only priority 1 was used.

Algorithm 1 performs traffic cleansing based on Snort, once the raw trace is loaded into memory, then, the type of alert to be used is read. In the case of alerts of type unified2, the maximum level of priority is read. Later, the while loop cycle reads the trace to be sanitized packet by packet to find a match with the alert file, if the packet was not related to an alert, the packet is written to the cleaned trace file. Otherwise, the priority level of the alert is checked and compared with max_priority, if the priority is higher than max_priority then the packet is written to the cleaned trace, trace_out file.

Table 1 shows the summary of UAN-12, our dataset of sanitized traces for this work, which is composed of 12 network traces. The total size is 47.8 GB and contains 78.7 million of TCP/UDP packets. All network traces were collected on weekdays at the same time of day to minimize errors derived from daytime effects on network use.
**Algorithm 1** Raw network dumps cleansing.  1:**procedure**NetworkClean  2:      load_trace_to_memory_buffer         ▹ network trace to be sanitized  3:      read_alert_logs pcap_logs or unified2_logs            ▹ snort alerts  4:      **if** pcap_logs **then**  5:          alerts_extraction(pcap_logs) snort_logs          ▹ map of snort alerts  6:      **if** unified2_logs **then**  7:          alerts_extraction(unified2_logs) snort_logs          ▹ map of snort alerts  8:          read_priority max_priority             ▹ severity level of the attack  9:      **while** extract_nexpacket(buffer) packet **do**10:          **if** is_bad_traffic (packet,snort_logs) **then**11:                **if** get_priority(packet,snort_logs) > max_priority **then**12:                  write_packet (packet) trace_out13:          **else**14:             write_packet (packet) trace_out;

### 3.2. Traffic Aggregation Phase

The objective of this phase is to generate a mixed network trace with attack-free traffic and malicious traffic. The malicious traffic dataset was created using NETRESEC [28] that offers a list of public repositories of PCAP files. Two resources from the list are malware-traffic-analysis [29] and the Stratosphere project [30] that offer public traces of network traffic in the presence of malware. In this work, a dataset from the Stratosphere IPS Project was used to represent malicious network traffic, specifically the CTU-13 dataset. From the aforementioned dataset, only the botnet network traffic traces of the twelve scenarios were used. Table 2 summarizes the malicious dataset used in this work.

Algorithm 2 shows the procedure for adding malicious network traffic to the sanitized trace. To do this, the duration of the sanitized trace duration_b is obtained, which, in this case, acts as background traffic on which malicious traffic will be mixed. The duration of the malicious traffic trace is also obtained since these measures will determine the time in which the attack will be launched, replay_delay_m. Then, a process is started to capture traffic on a network interface, ethernet_interface, e.g., eth0 for which the TCPDUMP tool will be used. After that, a process is executed for the replay of the background traffic, using tcpreplay in the same interface used for the capture. When the replay_delay_m time has arrived, the replay of the malicious trace begins. During the replay time of the background traffic, the traffic listened on the ethernet_interface is being captured. At the end of its replaying, the capture is interrupted and the mixed trace is obtained in mixed_traffic.

Table 3 summarizes the personalized network traffic traces that were generated by our framework, which consists of twelve network traffic traces made up of a mixture of normal and botnet traffic. Each trace represents a scenario where a specific botnet is spread. In the next phase, the 3Fex tool is used to extract the session features, label the sessions and generate a CSV file with this information.
**Algorithm 2** Malicious traffic aggregation by mixing network traces.  1:**procedure**MixingTraces  2:      duration_b ← get_the_capture_duration_in_seconds(background_traffic)  3:      duration_m ← get_the_capture_duration_in_seconds(malicious_traffic)  4:      replay_delay_m ← 0.5 * (duration_b - duration_m)  5:      start_traffic_capture(ethernet_interface,mixed_traffic)  6:      replay_traffic(background_traffic,ethernet_interface)  7:      wait(replay_delay_m)  8:      replay_traffic(malicious_traffic,ethernet_interface)  9:      **while** replay_trafifc(background_traffic) **do**10:          continue11:      stop_traffic_capture(ethernet_interface, mixed_traffic)

### 3.3. Feature Extraction and Labeling Phase

The session extraction and labeling phase are carried out using the 3Fex tool (Fast Flow Feature Extractor) which we develop in the C++ programming language. 3Fex is designed to handle network traces in libpcap 2.4 format and is efficient in extracting attributes from TCP/UDP sessions for IPv4 and IPv6. Table 4 shows the 52 session features that extracted our tool. Additionally, 3Fex has the option of generating an output file with the times between arrivals of the packets that make up each session. This allows other types of study, such as time series, heavy-tailed distribution, long-term dependence, or self-similarity to be performed.

Tools that extract session features from network traces, such as CICFlowMeter [31] or Flowmeter [32], have shown deficiencies when reconstructing network sessions since these do not extract all the packets that form the session. This behavior produces incorrect values in the features for that session. Another limitation is the long processing time to produce the feature file when processing large-size traces. In addition, none of them has the option to carry out traffic labeling.

Algorithm 3 shows the mechanism that our tool uses to handle network traces and extract the network traffic sessions and optionally carry out their labeling and/or attacks classification. To speed up the process and prevent access to the hard disk, the traffic trace is loaded into the RAM memory using a buffer. Once loaded into memory, the type of Snort alert to use, PCAP or unified2, is identified and a snort_logs map is created, which will function as an associative container that stores the information of the alerts in the form of a five-tuple and an object, the latter representing the alert attributes. Subsequently, a programming cycle allows to identify the network sessions and extract the set of selected features, features_session. Depending on the type of alert, for example, for PCAP alerts it is only possible to perform a binary labeling, i.e., 0, negative class, and 1 for positive class. In case of alerts based on unified2, we can additionally define the highest priority level to tag and/or the type of attack classification given by Snort, i.e., meeting the unified2_logs && snort_logs condition. Finally, a file generated in CSV format contains the features of the sessions and/or their labels and/or the classification of the type of attack. We select CSV files because it is a common data interchange file format used when working with open source languages, such as R or Python. Table 5 presents the summary of the sessions for the UAN-12 network dataset.
**Algorithm 3** Feature extraction and labeling from network traces.  1:**procedure**FeatureExtractionAndLabeling  2:     buffer ← load_trace_to_memory               ▹ network trace loaded in memory  3:     read_alert_logs pcap_logs or unified2_logs                       ▹ snort alerts  4:     **if** pcap_logs **then**  5:          snort_logs ← alerts_extraction(pcap_logs)               ▹ object of snort alerts  6:     **if** unified2_logs **then**  7:          snort_logs ← alerts_extraction(unified2_logs)               ▹ object of snort alerts  8:    **while** session ←extract_session(buffer) **do**  9:         features_session← feature_extraction(session)           ▹ extract packet features10:         **if** snort_logs **then** session_labeling(session,snort_logs, features_session)          ▹add label feature11:        **if** unified2_logs && snort_logs **then** session_classification(session,snort_logs,features_session)                    ▹ add classification feature    save_features_file(features_session, CSV_file)oad_trace_to_memory      ▹ write dataset

An outstanding feature is that, once a session has been extracted and processed, the search space in the trace is reduced since packets that were extracted in previous sessions are marked to be ignored in subsequent sessions.

The dataset UAN-12 generated by our platform is available at https://securitylab.uan.mx/dataset-uan12.htm (accessed on 30 December 2021). A virtual machine with a running version of the framework is available also at https://securitylab.uan.mx/dataset-uan12.htm (accessed on 30 December 2021). The source code is available at https://github.com/OliverITT/3FEx (accessed on 30 December 2021). Finally, the models source code for this paper is available at https://github.com/pvelardea/botnet-detection (accessed on 30 December 2021).

### 3.4. Classification Phase

Network traffic classification is a specialized solution and a valuable tool used to effectively tackle network planning, management, and monitoring, and also for attack detection and forensic analysis. Network attack detection can be accomplished through supervised ML.

ML algorithms learn from data. Specifically, in supervised ML, it is assumed that we have access to a dataset D assembled of *n* labeled, independent, and identically distributed training examples (i.i.d.), $\left(X_{1},Y_{1}\right),\ldots ,\left(X_{n},Y_{n}\right)$. Therefore, each instance or observation is a pair formed from a feature vector *X* belonging to a feature space or input space $X\in R^{p}$ together with the system’s result (class label), *Y* belonging to a label space or output space Y. The fundamental assumption of statistical learning theory is that there exists a defined joint probability distribution over the feature-label space X×Y, denoted as $P_{XY}\left(x,y\right)$, where X,Y denotes a pair of random variables distributed over $P_{XY}\left(x,y\right)$ and the pair (x,y) denotes a realization. The training dataset D is expressed as follows:(1)$$D={\left\{\left(x_{i},y_{i}\right)\right\}}_{i=1}^{n}\subset X\times Y=X\times \left\{0,1,\ldots ,k-1\right\}$$
where *k* is the number of classes.

Supervised ML is described as an approximation problem to a target function, y=f(x) that maps feature vectors $x_{i}$ with outputs $y_{i}$. Because the objective function is unknown, ML algorithms try to find a hypothesis function, h:X→Y that approximates f(x). The hypothesis function is also usually represented as h(x,θ), or $h_{\theta }\left(x\right)$, where θ is its parameters’ vector from the learned model from D. The set of all possible hypotheses is known as the hypothesis space, $H=\{h_{\theta }\left(.\right),\theta \in \Theta \}$, where Θ is the parameter space.

A learning algorithm is a procedure A:(X×Y)→H that takes the training set and produces the model that best approximates the unknown objective function, that is, $h_{\theta }\left(x\right)=A\left(D\right)=A\left(\left(X_{1},Y_{1}\right),\cdots ,\left(X_{n},Y_{n}\right)\right)$. Note that $h_{\theta }\left(x\right)$ is a function of random variables, so it is also a random variable.

The loss function, $L(h_{\theta }\left(x\right),y)$ measures the divergence or error, *e*, between the prediction made by the model and the correct value of the observation used during learning, resulting in a loss value:(2)$$L:X\times Y\times R\to [0,\infty^{+})$$

As an example, for k=2, in a binary classification, the loss function 0/1 is used frequently:(3)$$L\left(h_{\theta }\left(x_{i}\right),y_{i}\right)=1\left\{h_{\theta }\left(x_{i}\right)\neq y_{i}\right\}$$
where 1{A} is an indicative function that takes the value 1 if the logic condition *A* is true and 0 if not. *L* counts the number of mistaken classifications. Other loss functions for classification are: binary cross-entropy, Huber loss, ϵ-insensitive loss, hinge loss, logistic loss, exponential loss, among others.

The risk or generalization error for *h* is written as the expected value of the loss function
(4)$$R\left(h_{\theta }\right)=E_{XY}\left[L\left(h_{\theta }\left(x\right),y\right)\right]=\int L\left(h_{\theta }\left(x\right),y\right)dP_{XY}\left(x,y\right)$$
where the expectation is taken related to a distribution function $P_{XY}\left(x,y\right)$. The ideal estimator or objective function is the minimizer of:(5)$$\min_{h\in H}R\left(h_{\theta }\right)$$
where H is the hypothesis space in which $R(h_{\theta })$ is defined.

In practice, $h_{\theta }$ cannot be found in this way because $P_{XY}\left(x,y\right)$ is unknown. The only information available is in the training set, D. A natural estimator of the risk function is the empirical risk
(6)$$R_{emp}\left(h_{\theta }\right)=\frac{1}{n}\sum_{i=1}^{n}L\left(h_{\theta }\left(x_{i}\right),y_{i}\right)$$

Constructed from the training set, D. Learning $h_{\theta }$ by minimizing Equation (Equation 6) is known as the empirical risk minimization principle (ERM). ERM states that it is possible to minimize $R_{emp}\left(\Theta \right)$ with respect to Θ. For n→∞, the empirical risk, $R_{emp}\left(\Theta \right)$ converges uniformly to the risk function, R(Θ)[33,34].

Figure 4 shows how the concepts mentioned above are related to carrying out the learning of a supervised classification $h_{\theta }\left(x_{i}\right)$ model for a ML algorithm, A. Additionally, an input to the ML algorithm has been integrated to modify it and make it cost-sensitive strategy, which is effective to solve the unbalanced classification problem.

Class imbalance in binary datasets occurs when there exists a majority or negative class with normal data and a minority or positive class with abnormal or important data, which generally has the highest cost of erroneous classification [35]. One way to quantify the level of class imbalance is with the Imbalance Ratio (IR), which is the ratio of the number of instances of the negative class to the number of instances of the positive. For example, the distribution may be slightly skewed, a 4:6 IR, or severe, with an IR 1:100, 1:1000, or more. Traditional ML algorithms often assume that the training set is balanced. In unbalanced datasets, such as in intrusion detection datasets [36], classical classification algorithms can bias towards majority classes, and metrics, such as accuracy often give misleading values.

The introduction of cost-sensitive learning is necessary to remove the limitations of traditional classification algorithms for unbalanced datasets. Minority class oversampling and majority class subsampling can be used to handle this problem. When working with an unbalanced binary classification problem, the minority class (positive class) is usually the most significant interest. In our case, it would correspond to malicious traffic sessions, and as there are few samples, it is usually more difficult to predict. In this work, to evaluate our generated datasets, we use two approaches: data sampling and cost-sensitive algorithms.

Data sampling is a set of techniques that transform a set of training data to balance or improve the distribution between classes. Once balanced, traditional ML algorithms can be trained directly on the transformed dataset without modification. This technique can help to address the unbalanced classification problem. On the other hand, cost-sensitive learning considers the costs of prediction errors (and potentially other costs) when training a ML model. Therefore, instead of each instance being classified correctly or incorrectly, each class (or instance) receives a misclassification cost. Thus, rather than trying to optimize accuracy, the problem is to minimize the total cost of misclassifications [37].

## 4. Experiments and Results

In this paper, we conducted experiments with the dataset generated by our framework. The dataset comprises 12 scenarios that contain both legitimate activity and one type of botnet. For each scenario, the associate CSV file contains the traffic sessions with their corresponding 51 features plus the binary class label. Class 0 or negative corresponds to a benign network traffic session, while class 1 or positive corresponds to a botnet traffic session. The goal is to build a predictive model with a ML workflow in Python 3.8.5. The problem was addressed as an unbalanced binary classification since the distribution between the positive and negative classes is not uniformly distributed, presenting a severe bias towards the negative class, i.e., benign traffic sessions. The attack vector included in the dataset is a botnet network traffic. A botnet is a network of computers, called bots, which were infected with malware that allows them to be remotely controlled by an attacker, called a botmaster. These computers can be used together to carry out malicious activities without the owner’s knowledge [38]. The life cycle of a botnet comprises several steps, starting with the botmaster that infects the victim with malware. The infected bot connects to command and control (C&C) channels via HTTP, IRC, or other protocols. The botmaster sends orders to the bots through a C&C server and creates an army of bots little by little [39].

This article presents five types of classifiers based on ML algorithms to distinguish between benign and botnet network traffic sessions by classifying the corresponding session. A set of modified algorithms was used, i.e., cost-sensitive algorithms. Additionally, traditional ML algorithms are used accompanied by sampling techniques to balance the distribution of classes in the dataset.

The first classifier is W-LR which stands for Weighted-Logistic Regression, it is a cost-sensitive Logistic Regression algorithm. The second classifier is LR-SMOTE, the traditional logistic regression algorithm is applied together with one of the best known oversampling algorithms called SMOTE (Synthetic Minority Over-sampling Technique), which randomly generates synthetic objects between two objects of the minority class [40]. The third classifier is W-DT, a Weighted-Decision Tree, that is a cost-sensitive Decision Tree algorithm. The fourth classifier is SVM+OSS which is based on the Support Vector Machine (SVM) algorithm applied alongside with a subsampling technique called One-Sided Selection (OSS) [41], that combines Tomek Links and the Condensed Nearest Neighbor (CNN) Rule [42]. The last algorithm is XGB, extreme gradient boost better known as XGBoost which is implemented as an optimized distributed library under the gradient boosting framework designed to be highly efficient, flexible, and portable [43].

The ML pipeline used in this work is as follow:1.Split data into train and test sets;2.Fit data preparation on training dataset;3.Apply data transformation to obtain prepared train and test datasets;4.Find a model running grid search only on a prepared training set using cross-validation;5.Evaluate model on prepared test set using four performance metrics.

### 4.1. Performance Metrics

Typically, the following criteria are used to evaluate the performance of a ML predictive model in detecting botnet:1.True Positive (TP): indicates that the botnet was successfully detected in the traffic session.2.True Negative (TN): indicates that a benign traffic session was correctly identified.3.False Positive (FP): indicates that a benign traffic session was falsely detected as a botnet session.4.False Negative (FN): indicates that a botnet session was not detected and was identified as a benign traffic session.

Based on the previous criteria, different metrics in terms of macro-averages, specifically, macro-precision, macro-recall, and macro-F1 were used to assess the performance of the models proposed in this research. Recently, macro-averaging has been proposed in the literature to quantify the botnet detection rate, [38,44,45,46]. To define a macro-average we considered *m* as a performance measure for class *c*, which depends on the previous criteria. The precision measure for class c=0 and class c=1 are: $p_{0}=TN/\left(TN+FN\right)$ and $p_{1}=TP/\left(TP+FP\right)$, respectively. These measures are also known as Negative Predictive Value (NPV) and Positive Predictive Value (PPV), which are interpreted as the proportion of correct predictions regarding everything predicted for class *c*.

On the other hand, the recall for class 0 and class 1 are $r_{0}=TN/\left(TN+FP\right)$ and $r_{1}=TP/\left(TP+FN\right)$. These measures are also known as True Negative Rate (TNR) and True Positive Rate (TPR), which are interpreted as the proportion of correct predictions with respect to the size of each class *c*. The *F*1-scores for classes 0 and 1 are $f_{0}=2p_{0}r_{0}/\left(p_{0}+r_{0}\right)$ and $f1=2p_{1}r_{1}/\left(p_{1}+r_{1}\right)$, and correspond to the harmonic means, $p_{c}$ and $r_{c}$, respectively.

A macro-average metric counts the average for each independent class and then obtains a general average treating all classes as equals, including minorities. As mentioned in [47], the macro-averages of precision, recall and *F*1-score are more sensitive to the problem of class imbalance compared to their respective micro-averages.

Therefore, if we want to know the effectiveness in identifying the minority class, i.e., the malicious macro-averages must be calculated. A macro-average *M* of a measure $m_{c}$ can be calculated as follows:(7)$$M=\frac{1}{n}\sum_{c=0}^{n-1}m_{c}$$
where *n* is the total number of classes. In our case, this is a binary classification, n=2.

The macro averages for precision, recall, and *F*1-score are $P=1/2(p_{0}+p_{1})$, $R=1/2(r_{0}+r_{1})$, and $F1=1/2(f_{0}+f_{1})$, respectively. In this work, we also use the Matthews correlation coefficient (MCC), developed by Matthews in 1975 and proposed by Baldi et al. in 2000, as a standard performance metric for ML algorithms with a natural extension to the multiclass case [48]. Because it incorporates the four categories of a confusion matrix, several works consider the MCC metric more reliable than the *F*1 metric to assess binary classifiers with class imbalance [49,50]. However, this contrasts with the results in [51], where it is argued that *MCC* seriously deteriorates when a dataset is unbalanced. Under this scenario, we opted to consider macro-averages of precision, recall, *F*1-score, and *MCC* to compare the behavior of the classifiers under these metrics.

The MCC is defined as:(8)$$MCC=\frac{TP*TN-FP*FN}{\sqrt{\left(TP+FP)(TP+FN)(TN+FP)(TN+FN\right)}}$$

This metric has some properties: when the classifier is perfect (FP = FN = 0), the *MCC* value is 1, which indicates a perfect positive correlation. On the other hand, when the classifier always misclassifies (TP = TN = 0), the MCC value is −1, representing a perfect negative correlation (in this case, it is enough to invert the classifier’s result to obtain the ideal classifier). In fact, the MCC value is always between −1 and 1, and 0 means that the classifier is random.

### 4.2. Classification Algorithms Performance

Next, we show the experimental results using the twelve botnet attack scenarios represented by the dataset generated in this work. Five classification algorithms were used to perform the detection of botnet attacks. The chosen classifiers combine cost-sensitive algorithms and sampling techniques, adapting the ML methodology for the context of unbalanced data.

Figure 5 shows the first six scenarios, while Figure 6 shows the remaining ones. Each scenario is identified by a title that presents a set of bar graphs. These bar graphs show the performance of each classifier using the macro-averages of precision, recall, and *F*1-score, as well as the MCC, respectively. According to the results obtained in the UAN-12 dataset, it is observed that the macro average measures *F*1-score and MCC follow a congruent behavior between both by not presenting discrepancies in the evaluation results of the classifiers. In the comparison among the different methods selected to address the unbalanced classification problem, the SVM-OSS and XGB classifiers showed the highest performance versus W-DT, W-LR, and LR-SMOTE, which presented low performance in all scenarios, specially in scenarios 3, 4, 5, 9, and 11 where W-LR and LR-SMOTE performed the worst.

Table 6 summarizes the performance of the two best classifiers, SVM + OSS and XGB. Based on the results from macro average of F1 and MCC, both classifiers obtained an average performance of 0.97 and 0.94, respectively. Comparing the classifiers in each scenario, we can see that in scenarios 1, 3, and 7 the classifiers had the same performance. Meanwhile, SVM + OSS outperformed the rest by difference of hundredths in scenarios 4, 5, 8, 9, and 10. Similarly, XGB did so in scenarios 2, 6, 11, and 12. Regarding the computational processing times, the SVM + OSS classifier consumed 3.75 h for all the scenarios, considering only the undersampling times and the model adjustment. XGB, on the other hand, only required 0.45 hrs for model fitting. Is important to highlight that this classifier did not require applying any technique for class imbalance. The models generated for botnet detection were performed on a computer with an AMD Ryzen 9 3950X 16-Core Processor, 64 GB DDR4@ 3600 MHz and a NVMe M.2 SSDs using a PCIe 4.0 interface.

An interesting observation of this work was the behavior of the average macro metrics of F1 and MCC. In the 12 scenarios, these two metrics followed a correlated behavior. Figure 7 shows the variation of these measures for SVM + OSS and XGB classifiers that can be described by calculating the Pearson correlation coefficient of r = 0.99 in both classifiers, verifying a strong relationship between both metrics, concluding that both measures give consistent results.

The bibliography consulted in this research allowed us to identify that previous studies only made comparisons of the MCC measure against precision, recall, and *F*1-score, despite the three last measures, they only consider a subset of the dataset concerning the positive class. In contrast, we use macro averages for both classes, and the results shown that the performance of our classifiers was very similar in terms of *F*1-score and MCC measure.

## 5. Conclusions

The main goal of this work is to propose a framework for network datasets generation from real traffic traces that can be used in the research of NIDS. The generation of personalized datasets opens up new opportunities in NIDS research by providing dynamic scenarios that allow the representation of the most recent threats. In this sense, we can use network traffic threats that have been made publicly available or captured in honeypot networks.

Our proposed framework offers realistic scenarios, which were not available before. The traffic aggregation offers the advantage of using network traffic captures from our networks for later be used as background traffic combinable with malicious traffic of our interest. Another advantage is that the generated network traces offer full information, because these have not undergone any transformation process related to the anonymization of data that protect the privacy of the information. On the other hand, the tools developed during this research provide significant help by facilitating the reliable and efficient extraction of session features. Similarly, the traffic session labeling component contributes to the process of using supervised ML algorithms that require labeled data.

Finally, the usefulness of the proposed dataset generator framework was assessed using a generated unbalanced dataset. Five ML predictive models were used to detect botnet attacks. The performance of these algorithms was estimated using F1 macro-averaging-score and the MCC, identifying that these measures show a similar performance of classification with unbalanced data.

Some future work related with the proposed framework is focused in three aspects:1.Usability: Develop a GUI for the final user of the tool.2.Portability: Make use of container technology to easily deploy the tool in different environments.3.Performance: Use a preprocessing stage to handle larger amount of traffic, not limited by the RAM in the computer.

## Figures and Tables

**Figure 1 sensors-22-01847-f001:**
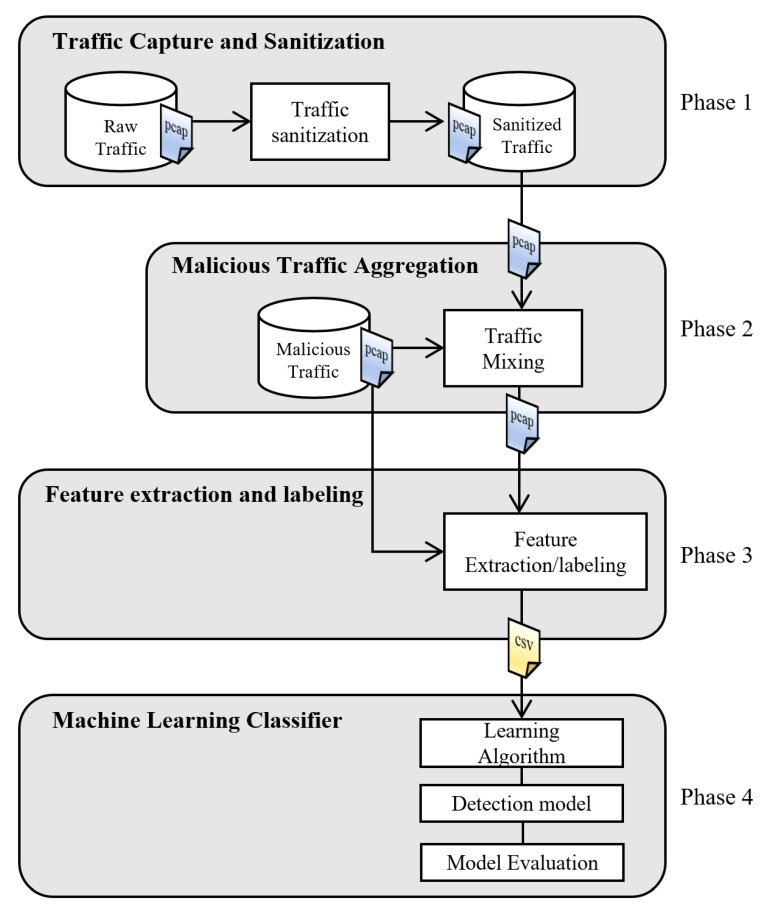
General diagram for the proposal approach.

**Figure 2 sensors-22-01847-f002:**
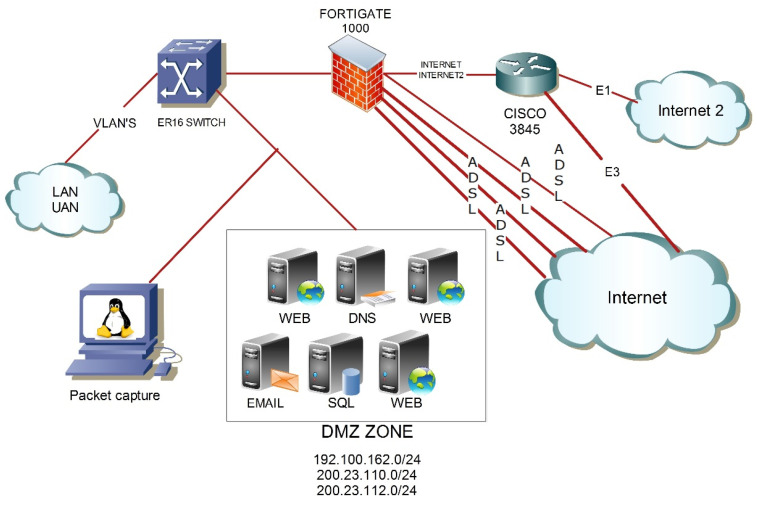
General diagram for the production network.

**Figure 3 sensors-22-01847-f003:**
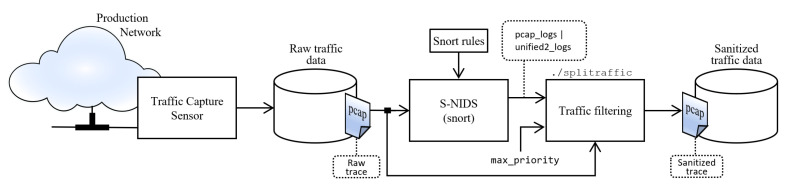
Diagram for the recollection and cleansing network traffic phases.

**Figure 4 sensors-22-01847-f004:**
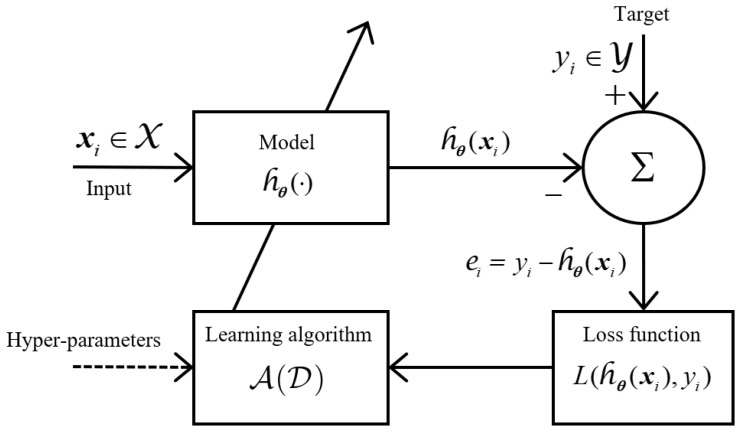
Schematic diagram for training a ML model.

**Figure 5 sensors-22-01847-f005:**
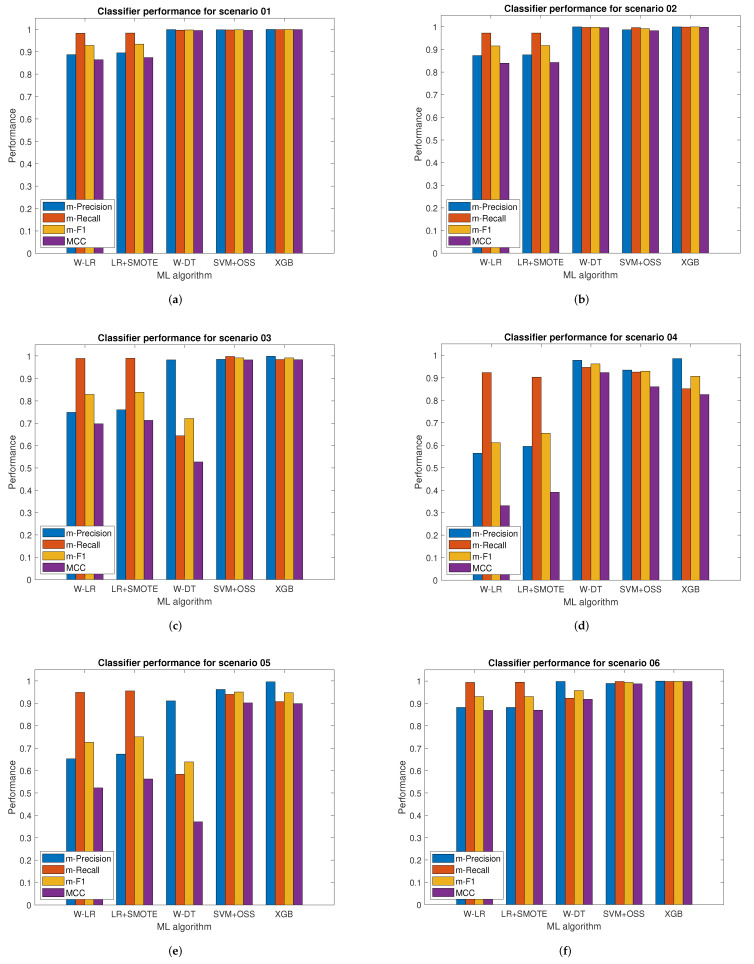
Classifiers performance in botnet attacks detection for scenarios 1 to 6. (**a**) Scenario 1. (**b**) Scenario 2. (**c**) Scenario 3. (**d**) Scenario 4. (**e**) Scenario 5. (**f**) Scenario 6.

**Figure 6 sensors-22-01847-f006:**
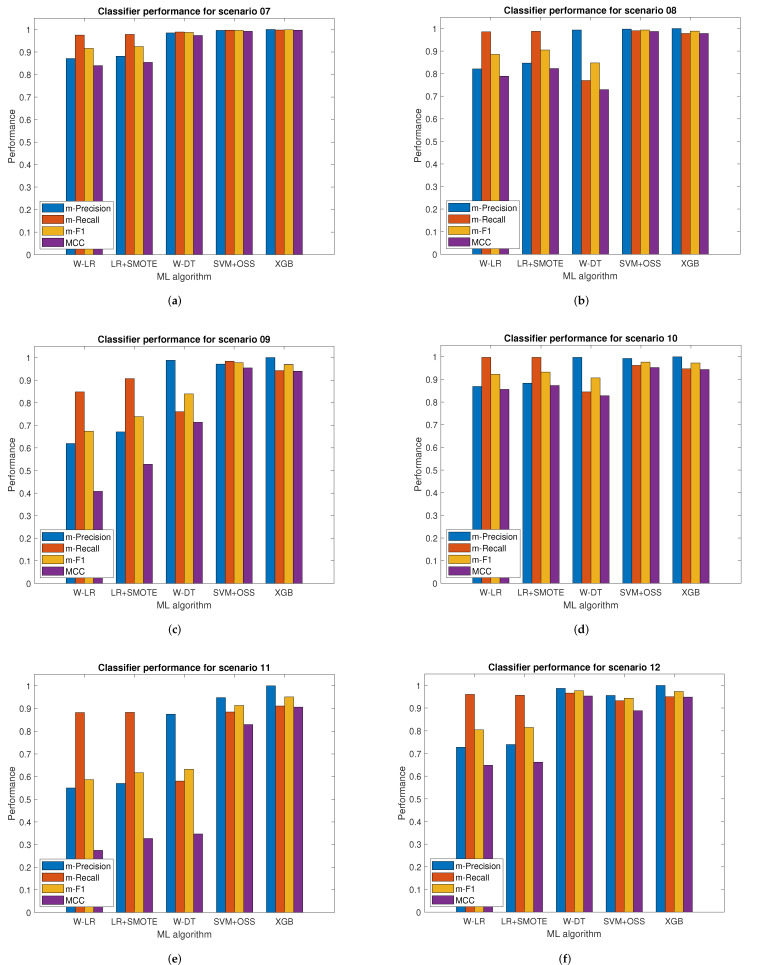
Classifiers performance in botnet attacks detection for scenarios 7 to 12. (**a**) Scenario 7. (**b**) Scenario 8. (**c**) Scenario 9. (**d**) Scenario 10. (**e**) Scenario 11. (**f**) Scenario 12.

**Figure 7 sensors-22-01847-f007:**
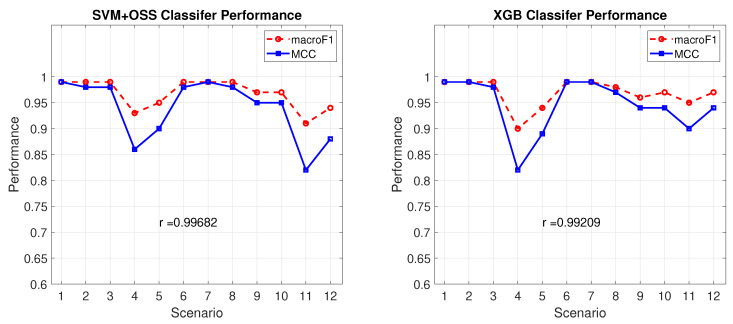
Classifiers performance graphs and Pearson Correlation Coefficient, *r*.

**Table 1 sensors-22-01847-t001:** Cleansed network dumps from UAN-12 dataset.

Name of File	Number of Packets (M)	Size (GB)	Duration Time (s)
120426070000.pcap	6.02	3.6	28,799.32
120427070000.pcap	5.02	2.7	28,798.68
120509070000.pcap	5.54	3.2	28,799.59
120511070000.pcap	5.01	2.9	28,799.76
120430070000.pcap	4.45	2.6	28,799.61
120516070000.pcap	6.97	3.9	28,799.20
120502070000.pcap	4.48	2.2	28,799.49
120529070000.pcap	5.26	2.9	28,800.40
120604070000.pcap	8.97	6.3	28,799.91
120709070000.pcap	10.00	7.2	24,886.23
120706070000.pcap	11.00	6.5	28,798.97
120602070000.pcap	5.94	3.8	28,799.03

**Table 2 sensors-22-01847-t002:** Summary of the malicious network traces.

Trace Name	Packets (K)	Size (MB)	Duration Time (s)	Bot	No.
botnet-capture-20110810-neris.pcap	322	55.1	17,106.04	Neris	1
botnet-capture-20110811-neris.pcap	175	34.5	12,263.16	Neris	2
botnet-capture-20110812-rbot.pcap	26	2.5	25,142.94	Rbot	3
botnet-capture-20110815-rbot-dos.pcap	256	212.1	3890.61	Rbot	4
botnet-capture-20110815-fast-flux.pcap	45	29.5	1213.34	Virut	5
botnet-capture-20110816-donbot.pcap	24	5	7217.54	Menti	6
botnet-capture-20110816-sogou.pcap	20	18	873.32	Sogou	7
botnet-capture-20110816-qvod.pcap	34	9.2	25,193.67	Murlo	8
botnet-capture-20110817-bot.pcap	2124	1000	10,002.54	Neris	9
botnet-capture-20110818-bot.pcap	62,000	8300	17,397.17	Rbot	10
botnet-capture-20110818-bot-2.pcap	3941	4000	477.52	Rbot	11
botnet-capture-20110819-bot.pcap	351	281.2	3796.01	NSYS.ay	12

**Table 3 sensors-22-01847-t003:** Summary of the UAN-12 dataset.

File Name	Packets (M)	Size (GB)	Duration Time (s)
esc-01-Mixed-traffic.pcap	6.34	3.61	2816.57
esc-02-Mixed-traffic.pcap	5.19	2.75	28,813.03
esc-03-Mixed-traffic.pcap	5.57	3.24	2820.25
esc-04-Mixed-traffic.pcap	5.27	3.08	28,814.75
esc-05-Mixed-traffic.pcap	4.5	2.63	28,812.05
esc-06-Mixed-traffic.pcap	6.99	3.91	28,817.06
esc-07-Mixed-traffic.pcap	4.51	2.21	28,812.28
esc-08-Mixed-traffic.pcap	5.29	2.93	28,819.35
esc-09-Mixed-traffic.pcap	9.05	6.29	28,822.44
esc-10-Mixed-traffic.pcap	77	15.4	24,911.77
esc-11-Mixed-traffic.pcap	15	10.4	28,835.23
esc-12-Mixed-traffic.pcap	5.99	3.79	28,820.96

**Table 4 sensors-22-01847-t004:** Network sessions proposed features.

No.	Feature Name	Description
1	protocol	Protocol type
2	Ts	Session time stamp
3	ip_shost	IPv4 source address
4	ip_dhost	IPv4 destination address
5	ipv6_shost	IPv6 source address
6	ipv6_dhost	IPv6 destination address
7	srcPort	Source port
8	dstPort	Destination port
9	fduration	Session duration
10	total_fpackets	Number of packets in the forward direction
11	total_bpackets	Number of packets in the backward direction
12	total_fpktl	Transmited bytes in forward direction
13	total_bpktl	Transmited bytes in backward direction
14	min_fpktl	Minimum packet size in forward direction
15	min_bpktl	Minimum packet size in backward direction
16	max_fpktl	Maximum packet size in forward direction
17	max_bpktl	Maximum packet size in backward direction
18	mean_fpktl	Mean packet size in forward direction
19	mean_bpktl	Mean packet size in backward direction
20	std_fpktl	Standard deviation packet size in forward direction
21	std_bpktl	Standard deviation packet size in backward direction
22	total_fipt	Sum of interarrival times in forward direction
23	total_bipt	Sum of interarrival times in backward direction
24	min_fipt	Minimum interarrival time in forward direction
25	min_bipt	Minimum interarrival time in backward direction
26	max_fipt	Maximum interarrival time in forward direction
27	max_bipt	Maximum interarrival time in backward direction
28	mean_fipt	Mean interarrival time in forward direction
29	mean_bipt	Mean interarrival time in backward direction
30	std_fipt	Standard deviation packet size in forward direction
31	std_bipt	Standard deviation packet size in backward direction
32	fpsh_cnt	Number PSH flags in forward direction
33	bpsh_cnt	Number PSH flags in backward direction
34	furg_cnt	Number URG flags in forward direction
35	burg_cnt	Number URG flags in backward direction
36	total_fhlen	Sum of headers length in forward direction
37	total_bhlen	Sum of headers length in backward direction
38	fPktsPerSecond	Packets per second in forward direction
39	bPktsPerSecond	Packets per second in backward direction
40	flowBytesPerSecond	Total Bytes per second in session
41	mean_flowpktl	Average package length
42	std_flowpktl	Standard deviation of package length
43	mean_flowipt	Average of interarrival times
44	std_flowipt	Standard deviation of interarrival times
45	flow_fin	Number of packets with FIN flag set
46	flow_syn	Number of packets with SYN flag set
47	flow_rst	Number of packets with RST flag set
48	flow_ack	Number of packets with ACK flag set
49	flow_urg	Number of packets with URG flag set
50	flow_cwr	Number of packets with CWR flag set
51	flow_ece	Number of packets with ECE flag set
52	downUpRatio	Download and upload ratio

**Table 5 sensors-22-01847-t005:** Summary of the sessions on UAN-12 dataset.

File Name	Size (MB)	Benign Sessions	Malicious Sessions	Ratio Imbalance
esc-01-Mixed-traffic.csv	102.0	292,386	18,277	1:16
esc-02-Mixed-traffic.csv	88.8	289,490	18,264	1:16
esc-03-Mixed-traffic.csv	140.0	420,074	2704	1:160
esc-04-Mixed-traffic.csv	100.0	303,587	197	1:1500
esc-05-Mixed-traffic.csv	77.7	232,174	872	1:250
esc-06-Mixed-traffic.csv	128.0	382,823	4567	1:100
esc-07-Mixed-traffic.csv	101.0	303,081	50	1:6050
esc-08-Mixed-traffic.csv	114.0	339,416	3341	1:100
esc-09-Mixed-traffic.csv	118.0	352,407	3465	1:100
esc-10-Mixed-traffic.csv	142.0	430,467	4606	1:100
esc-11-Mixed-traffic.csv	183.0	553,173	257	1:2100
esc-12-Mixed-traffic.csv	68.7	206,526	631	1:325

**Table 6 sensors-22-01847-t006:** Classifiers performance summary.

		SVM + OSS		XGB	
	**Metric**	**m-F1**	**MCC**	**m-F1**	**MCC**
**Scenario**					
1		0.99	0.99	0.99	0.99
2		0.99	0.98	0.99	0.99
3		0.99	0.98	0.99	0.98
4		0.93	0.86	0.90	0.82
5		0.95	0.90	0.94	0.89
6		0.99	0.98	0.99	0.99
7		0.99	0.99	0.99	0.99
8		0.99	0.98	0.98	0.97
9		0.97	0.95	0.96	0.94
10		0.97	0.95	0.97	0.94
11		0.91	0.82	0.95	0.90
12		0.94	0.88	0.97	0.94
Overall performance		0.97	0.94	0.97	0.94

## Data Availability

The dataset UAN-12 generated by our platform is available at https://securitylab.uan.mx/dataset-uan12.htm (accessed on 30 December 2021). A virtual machine with a running version of the platform is available also at https://securitylab.uan.mx/dataset-uan12.htm (accessed on 30 December 2021). The source code of the platform is available at https://github.com/OliverITT/3FEx (accessed on 30 December 2021). Finally, the models source code for this paper is available at https://github.com/pvelardea/botnet-detection (accessed on 30 December 2021).

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
