# Peer review of "A Novel Framework for Generating Personalized Network Datasets for NIDS Based on Traffic Aggregation"

_sensors, 2022, doi:10.3390/s22051847_

Round 1
Reviewer 1 Report
This paper presented a method for Generating Personalized Network
Datasets and Traffic Classification.
major issues:
I read the source code of the paper and find that there are some major issues:
use "esc-01-noIPs-paper.ipynb" as an example
(1) Cell [33] Feature selection
This section used both the training and test data for feature selection, which is wrong. Test data should never be used for feature and model selection.
(2) Grid search with Logistic Regression
Again, it used both the training and test data for feature selection, which is wrong. Test data should never be used for feature and model selection.
(3) To compare different models, there should be a unique test dataset.
Two standard approaches for a machine learning study:
(1) cross-validation based
split the whole data into train and test sets
run grid search only on training set using cross-validation
once the best model is found, evaluate the model on the test set
(2) train-validation based
split the whole dataset into a train set and a test set
split the train set into a "pure" train set and a validation set
run grid search on the "pure" training set, and measure the performance on the validation set. The best model hast the best performance on the validation set.
once the best model is found, evaluate the model on the test set
mini issue:
line 42 "Recently, new dynamic and synthetic datasets based on specific profiles have been developed" please cite the papers of these datasets
Reviewer 2 Report
I have the following recommendations regarding improvements of the paper.
1. Add the motivation of the proposed approach.
2. Literature related to the proposed scheme published in the last 3 years should be added (at least 3 more references).
3. Write the Research gap i.e. which mentions the limitations in the literature and how the authors overcome it.
4. Problem statements need to be written that clearly illustrate which problem is the focus of the study.
5. Research contributions, which clearly describe the novelty of the proposed solution should be added in the form of bullets or numbered form (1, 2, 3, ...).
6. Mention some future directions and limitations of the proposed scheme in the conclusions section.
7. Mention all the acronyms in a table by the end of the manuscript.
8. Paper should be checked for english spellings/minor grammatical mistakes.
9. Improve the figures with different fill/colors to clearly distinguish one scheme from the others.
Reviewer 3 Report
The authors have performed a very nice study of a network intrusion detection problem when the available data for the analysis is scarce. This is a quite relevant issue and should be published. However, I recommend the authors to consider the points raised (see list below) and carefully revise their manuscript.
Minor points:
(1) Table 4 should appear before Algorithm 3, because it is quoted in that order. However, this can be done at the editorial offices.
(2) Table 6 is quoted in the text much after it is shown. Please either move Table 6 down to the text (where it is quoted), or quote Table 6 before it appears. It is difficult to read in this way.
(3) Line 63: Maybe you use the notation 100MB if you refer to megabites.
(4) Line 91: 'The rest of the paper is...'
(5) Line 168: 'the' instead of 'teh'.
(6) Line 302: '...to a target function,...'
(7) Line 307: a closing parenthesis is missing for 'H={...}'
(8) Line 312: insert a comma after e: '...or error, e, between...'
(9) Line 319: you can join line 319 to line 318, i.e. no break, which looks nicer, keeping the sense of the sentence.
(10) Line 420: are p_c and r_c the harmonic means of p_i and r_i?
If yes, then write: '...to the harmonic means, p_c and r_c, respectively.'
Please define them explicitly to avoid confusion.
(11) Caption to Fig.7: Add the parameter r to the caption.
For instance: '...Pearson Correlation Coefficient, r'.
(12) Line 471: change 'obervate' into 'observe'.
(13) A list of abbreviations used should be provided at the end of the paper for convenience of the reader.
References to be added to Section 2:
D.Q. Le, T. Jeong, H.E. Roman, and J.W.K. Hong. 2011.
Traffic dispersion graph based anomaly detection.
In Proceedings of the Second Symposium on Information and
Communication Technology (SoICT '11).
Association for Computing Machinery, New York, NY, USA, 36–41.
DOI: https://doi.org/10.1145/2069216.2069227
which is related to DoS attacks and how to detect them. It would
complement the discussion in Sect.2.
Round 2
Reviewer 1 Report
good revision